# Microbiome and MicroRNA or Long Non-Coding RNA—Two Modern Approaches to Understanding Pancreatic Ductal Adenocarcinoma

**DOI:** 10.3390/jcm12175643

**Published:** 2023-08-30

**Authors:** Wiktoria Maria Izdebska, Jaroslaw Daniluk, Jacek Niklinski

**Affiliations:** 1Department of Gastroenterology and Internal Medicine, Medical University of Bialystok, 15-089 Bialystok, Poland; 2Department of Clinical Molecular Biology, Medical University of Bialystok, 15-089 Bialystok, Poland

**Keywords:** pancreatic ductal adenocarcinoma, microRNA, long non-coding RNA, microbiome

## Abstract

Pancreatic ductal adenocarcinoma (PDAC) is one of humans’ most common and fatal neoplasms. Nowadays, a number of PDAC studies are being conducted in two different fields: non-coding RNA (especially microRNA and long non-coding RNA) and microbiota. It has been recently discovered that not only does miRNA affect particular bacteria in the gut microbiome that can promote carcinogenesis in the pancreas, but the microbiome also has a visible impact on the miRNA. This suggests that it is possible to use the combined impact of the microbiome and noncoding RNA to suppress the development of PDAC. Nevertheless, insufficient research has focused on bounding both approaches to the diagnosis, treatment, and prevention of pancreatic ductal adenocarcinoma. In this article, we summarize the recent literature on the molecular basis of carcinogenesis in the pancreas, the two-sided impact of particular types of non-coding RNA and the pancreatic cancer microbiome, and possible medical implications of the discovered phenomenon.

## 1. Introduction

Pancreatic ductal adenocarcinoma (PDAC) is one of the most common and fatal neoplasms in humans [1]. It is estimated that PDAC will be the second leading cause of cancer death in the United States of America by 2030 [2]. The high mortality of PDAC is due to its late detection caused by a lack of sufficient and specific biomarkers and resistance to treatment [3]. The overall survival rate of PDAC is 9% [4,5]. The vast majority of patients are not suitable for surgical treatment and undergo chemotherapy and palliative therapy instead.

According to Wang et al., the human microbiota consists of microbes colonizing different sites of the human body as well as its ecosystem [6]. The term microbiome refers to the genetic information encoded by microbiota and its host. Microbiota consists of bacteria, fungi, viruses, and archaea. It has been discovered that the pancreas has its own microbiota, which is more abundant in a cancerous pancreas compared to a healthy pancreas [7,8]. The microbiota of a cancerous pancreas consists mostly of *Proteobacteria*, *Bacteroides*, and *Firmicutes* and is different from the one in a healthy pancreas [9]. Recent studies have shown that the microbiome of the pancreas and other organs contributes to shaping the course and progression of PDAC [1]. The pancreatic microbiome is partially predestined to treatment response and impacts the tumor environment [10].

MicroRNA (miRNA) and long non-coding RNA (lncRNA) are subtypes of non-coding RNA, which are functional RNA molecules without protein-coding abilities. miRNA consists of approximately 22 nucleotides, processed from larger hairpin precursors. As Mahesh et al. has stated, approximately 60% of all protein-coding genes are regulated by miRNAs with multiple targets per miRNA type [5]. LncRNA are longer than 200 nucleotides [11]. A significant impact of non-coding RNA on PDAC progression but also in repression via different pathways has already been discovered [12]. Both miRNA and lncRNA affect the carcinogenesis of PDAC by binding to different genes and one another, and thus modulate the production of amino acids of different metabolic pathways. They regulate numerous actions of carcinogenesis such as proliferation, cell differentiation, cell apoptosis, migration, and the creation of metastases [13,14].

The modern work of Wang et al. suggests a bi-directional influence of miRNA and bacteria in the gut microbiome on one another, which can promote carcinogenesis in the pancreas, but the microbiome also has a visible impact on the miRNA [7]. This implies that it is possible to use a combined impact of the microbiome and noncoding RNA to suppress the development of PDAC.

In this article, we summarize the recent literature on the molecular basis of carcinogenesis in the pancreas, the impact of particular types of non-coding RNA and the pancreatic cancer microbiome, and possible medical implications of the discovered phenomenon.

## 2. Evolution of Pancreatic Adenocarcinoma—What Occurs in Cells before Cancer?

Several risk factors predispose to the development of PDAC, including smoking, obesity, diabetes, chronic pancreatitis, and bacterial or viral infections (*Helicobacter pylori*, Hepatitis B Virus, Hepatitis C Virus) [3]. Another important factor is the acquisition of many genetic mutations, such as KRAS (85–100% of pancreatic cancer cases) and mutations of suppressor genes such as p16 (CDKN2A), TP53, SMAD, and mutations of damage repair genes like hMLH1 [6]. The accumulation of genetic mutations over time combined with exposure to environmental risk factors or familial predisposition promotes cellular damage and the development of precancerous lesions of PDAC. The most important risk factors are shown in Figure 1. 

The precursor pre-cancer state of PDAC is pancreatic intraepithelial neoplasia (PanIN), whilst states predisposing to cancer are intraductal papillary mucinous neoplasms (IPMNs) and mucinous cystic neoplasms (MCNs).

The graph above shows the most common risk factors leading to pancreatic ductal adenocarcinoma (based on the work by Ryan et al.) [3].

## 3. Genetic Alterations in PanIN, MCN, and IPMNs

The early benign lesions preceding PDAC are pancreatic intraepithelial neoplasia (PanIN), intraductal papillary mucinous neoplasm (IPMN), and mucinous cystic neoplasm (MCN). PanINs are microscopic changes, and MCN and IPMN are macroscopic lesions visible in imaging diagnostics [15].

The PanINs can be classified into three grades, PanIN-I, -II, and –III, based on their cytology, architecture, and nuclear characteristics, with grade I being the least advanced and grade III being referred to as “carcinoma in situ” [16]. One of the earliest genetic changes in PanIN is loss of telomeric integrity, which accounts for a significant majority of these type of lesions [17]. A mutation present in both mucinous lesions is Kirsten Rat Sarcoma (KRAS), detected in 90% of invasive PanINs [18] and the vast majority of IPMN [15,19,20]. The KRAS mutation causes impaired GTPase activity of KRAS2 products, which results in a constitutively active protein that is involved in the signaling transduction. Another frequent mutation in PanIN is tumor suppressor gene CDK2/INK4A, causing loss of p16, which inhibits the G1-S cell transition. The mechanism of these genetic alterations is due to various intragenic mutations, deletions, and promoter methylation [21,22]. In grades 2 and 3 of PanIN, mutations of TP53 (causing impaired ability to respond to stress in physiological processes) [15,23] and tumor suppressor gene BRCA2 [24], as well as the mutation of MADH4/SMAD/DPC4 (responsible for creating tumor suppressor protein Dpc4, causing insufficient gene expression regulation) [25,26], are present. Additionally, it has been reported that in PanIN, as well as in invasive adenocarcinomas, carcinogenetic proteins created by mutated genes are overexpressed. Some of these are components of epidermal growth factors, Notch [27], and Hedgehog [28] signaling pathways.

Intraductal papillary mucinous neoplasm (IPMN) is a lesion that produces mucin and arises from major or secondary pancreatic ducts [29]. The most common mutations in this lesion are KRAS, GNAS, and RNF43 [30]. The MADH4/SMAD/DPC4 mutation also appears in IPMN, yet only in the most advanced lesions [15]. Another common characteristic of IPMN is the expression of apomucins such as MUC1, MUC2, and MUC3 [31]. Other genetic alterations found in IPMN are the STK11/LKB1 mutation (perceived in one-third of cases, which results in the impaired production of serine/threonine protein kinase, also present in patients with Peutz–Jeghers Syndrome [32]). The mutation of GNAS occurs in around 4% of IPMNs. It is a secondary mutation activated by the mutated KRAS gene and affects the subnormal Gα_s_ protein by activating the adenylyl cyclase cascade of G-protein-coupled receptors, therefore responding to hormones and other extracellular signals) [3,15,33]. An amount of 15% of IPMN has the RNF43 mutation, impairing the antagonist of Wnt signaling [3].

Mucinous Cystic Neoplasms (MCNs) are mucin-producing neoplasms with distinctive ovarian-type stroma, present in the epithelium of the body and tail of the pancreas. The female gender predisposes to this neoplasm as over 90% of patients suffering from it are middle-aged women [34]. The MCNs can be classified based on histological architecture into three stages: mild MCN—mucinous cystadenomas, MCN with moderate dysplasia (MCN borderline), and MCN with carcinoma in situ [34]. Analogically to PanIN, one of the earliest molecular changes in MCN development is the KRAS mutation, present in sparse lesions of the mild stage, yet in up to all high-grade dysplasias [35]. Moreover, the advanced lesions are characterized by the inactivation of TP53 and MADH4/MAD4/DPC4 and mutation of RNF43 [16,36].

Table 1 compares the most common mutations in three early benign lesions proceeding to PDAC (Chr. stands for chromosome). It is based on the work of Singh et al. 2007 [15].

## 4. Genetic Alterations in PDAC and Their Consequences

PDAC is characterized by severe metabolic stress caused by hypovascularization, leading to extreme hypoxia and limitations in nutrition.

All of the factors causing metabolic stress and therefore limitations in nutrition create conditions for genetic mutations in cancer cells. As the gene mutations accumulate, the precancerous lesions may turn into pancreatic adenocarcinoma. Genetic modifications in PDAC may be divided into epigenetic regulatory gene mutations, histone modification enzyme mutations, chromatic regulating gene and cell proliferation gene mutations, and epigenetic differentiation such as histone-modifying genes [6].

The most common and one of the most researched mutations is the KRAS mutation. It is present in over 90% of PDAC according to the CGD (CITIZEN-GENERATED DATA) Data Portal by the American National Institutes of Health. This mutation plays a significant role in cancer initiation and maintenance. Within the RAS pathway, it can activate the effector signaling cascades and transcription factors that are involved in cell proliferation, transformation, and metastasis. Activation of the RAS pathway promotes pro-inflammatory signaling by activating NF-kappa beta, which, in PDAC, is considered to moderate many aspects of cancer development and progression [37]. Tumors with KRAS have constitutively high levels of autophagy [38], which can lead to both the detoxification of the cell from damaged particles and the provision of metabolites for biosynthesis and energy production [3,6]. It has been discovered in both mouse models and clinical trials that the progression of PDAC is inhibited by genetic inhibition of autophagic processes or by chloroquine (as it inhibits lysosomal acidification) [38]. KRAS also alters the expression of enzymes involved in glucose utilization. PDAC demands a high level of glycolysis, instead of oxidative phosphorylation [39]. Mutant KRAS promotes mitochondrial translocation of phosphoglycerate kinase-1. That results in the production of phosphorylated pyruvate dehydrogenase kinase-1 and restricted oxidative phosphorylation (OXPHOS) [40]. The KRAS mutation is involved in most of the metabolic reprogramming pathways that occur in pancreatic cancer. Therefore, it is crucial to incorporate KRAS mutation silencing or repairing as a therapeutic target.

In addition to the KRAS mutation, PDAC is abundant in hundreds of different mutations and the ones characterized below are the most common. The SMAD family member 4 (SMAD4) mutation is caused by homozygous deletion or mutation. It lowers the amount of SMAD-4-dependent inhibition of transforming growth factor-beta (TGF-beta) and reduces the induction of TGF-beta signaling, which promotes pro-tumorigenic responses [41]. The cyclin-dependent kinase inhibitor 2A (CDKN2A), loss of p16, appears in around half of PDAC cases [6]. It may lead to the loss of regulation of the cyclin-dependent kinase (CDK) 4 and 6 cell cycle checkpoints, which cause the dysregulation of the cell cycle and subsequent carcinogenesis [42]. The inactivation of TP53 is detected in 50 to 74% of PDAC cases according to Wang et al. [6]. Combined with the KRAS mutation, it can promote metabolic changes before malignant transformations. 

Other mutations found in less than 7% of PDAC cases according to the Harmonized Cancer Data Portal created by the National Institutes of Health are MUC16 (6,7%), ring-finger protein 47 (RFN43) (5,95%), GLI3 (5,59%), TGFBR (4,32%), and GNAS (3,91%). Another type of mutation detected in 10 to 25% of PDAC lesions is loss-of-function mutations encoding components of the SWI/SNF nucleosome remodeling complex [43]. 

The stem cell state is crucial in the development of PDAC. Cancer stem cells (CSCs) are cells of any state and purpose. Due to their plasticity, they can transform into the stem state at any time and renew infinitely [44]. This phenomenon explains cancer development and survival. The CSCs were first identified by the expression of CD44, CD24, and epithelial-specific antigen (ESA) [45]. It is known that CD133+/CXCR4+ cells (cell with both CD133 and CXCR4 antigens present, both being cancer stem cell markers) are responsible for metastasis development [46]. Other studies suggest that ALDH+ cells have CSC properties in pancreatic cancer [47]. ALDH+ cells are more tumorigenic than CD133+. Additionally, ALDH++/CD44+/CD24− cells are also highly tumorigenic [48].

Amidst all researched genes, FoxQ1 seemed to be vital for the aggressive biological properties of these cells. It knocks down EpCAM (epithelial cell adhesion molecule), which is overexpressed in malignant tumors [49]. Moreover, FOXQ1 restrains the Snail gene, which regulates the epithelial-to-mesenchymal transition of cells in cancer-stem-cell-like cells [50,51]. Taken together, it seems crucial to incorporate CSC-targeted treatment in the pancreatic treatment protocol [44]. A promising method to improve chemotherapy could be the double inhibition of the sonic hedgehog pathway and mTOR pathway as they significantly eradicate pancreatic CSCs and result in a long survival rate in both mice and human xenograft models [52,53].

Considering the general trend of focusing on high-throughput studies, more and more research using omics technologies emerge in the pre-cancer lesions and PDAC. As it can give a deeper understating of processes underlying the progression and development of PDAC [54,55,56], most of the omics studies focus on finding a perfectly sensitive biomarker of PDAC [57]. A novel work of Kobayashi et al. summed and compared the proposed biomarkers for PDAC, including germline mutations, miRNA, metabolites, and sugars [57]. Other works focus on finding the best way to differentiate and group subtypes of PDAC based on their histology and gene expression [58]. Yet, it is quintessential to realize that the majority of genomic analysis is performed on resected lesions, which is available only in around 15% of PDAC cases. [3]. Omics research sheds new light on the complexity of pancreatic cancer and its origin. Therefore, it is crucial to conduct omics research on the non-resective cases of pancreatic adenocarcinoma such as analyzing biopsy material.

## 5. Molecular Aspects of PDAC Metastases

Cancer metastases are formed by a detachment of cancer cells from the primary mass and transport via blood or lymphatic vessels. The majority of pancreatic metastases are localized in the liver (79–94%), peritoneum (41–56%), abdominal lymph nodes (41%), and the lungs (45–48%) [59,60,61,62]. According to Oldfield et al., all patients with pancreatic cancer have cells with metastatic potential in their primary lesion at the time of diagnosis [63]. The main difference between primary and metastatic tumor cells is the ability to proliferate, which is variable in the metastases as it is greatly influenced by the tumor environment [63].

Against popular belief that metastases occur only in the late [64,65] stage of cancer, in pancreatic cancer, they can appear even before the large mass formation of the primary lesion. It is mostly plausible that the step-by-step accumulation of different mutations and epigenetic alterations is the impulse for this process [66]. The so-far-discovered driving forces for metastases in pancreatic cancer are numerous non-coding RNAs [67,68], transcription factors (like Kruppel-like factor 4, KLF4 [69]), growth factors (like vascular endothelial growth factor, VEGF [70]), and oxygen conditions. MicroRNA is associated with a high risk of lymph node metastasis in patients with pancreatic cancer [71].

An experimentally reduced metastatic behavior of tumor has been observed by reducing the transcription of RUNX3 factor, which is overexpressed in KRAS and TP53 mutant mice (KPC mice). The floxed SMAD4 gene leads to rendering the mutational status of the TP53 allele in the above-described mice. Since the SMAD gene regulates the RUNX3 levels, the reduction in metastasis can be achieved in this way [72].

A possibly relevant way to understand and manage pancreatic cancer metastases is to explore the epithelial-to-mesenchymal (ETM) transition in PDAC. The EMT-transcription factor Zeb1 is responsible for enabling the colonization and plasticity of the phenotype of tumor cells and promotes PDAC metastasis [73]. Reducing Zeb1 suppresses the colonization of cancer and the phenotypic/metabolic plasticity of cancerous cells [74]. Therefore, high Zeb1 levels in PDAC correlate with poor diagnosis [74].

Another crucial aspect of PDAC metastasis is altered genes. Based on the organoid culture system, Roe et al. discovered that the metastatic process goes hand in hand with recurrent alterations in enhancer activation. Therefore, it adds to the likelihood of the transcription of genes responsible for PDAC activity and its aggressiveness [75]. Enhancers are DNA sequences that increase the transcription of particular genes [76]. The main driver of enhancer activation in PDAC metastasis is FOXA1. It makes the cells more invasive and more predestined to metastasize in vivo, and it makes the anchorage less dependent on in vitro growth [75].

Table 2 presents the most common genetic mutations in the particular stages of carcinogenesis in the pancreas causing PDAC and metastases, as well as characteristic mutations present in the pre-malignant lesions observed prior to PDAC.

## 6. miRNA and lncRNA Mechanisms of Intervention in PDAC Progression

Both microRNA and long non-coding RNA belong to the family of non-coding RNA. They are transcripts without the function of coding proteins. They can be divided by size (short and long ncRNA), function (housekeeping RNA and regulatory RNA), direction of transcription (sense/antisense, bidirectional), as well as location (intronic and intergenic). Non-coding RNA may interact with a wide range of other molecules such as messenger RNA (mRNA) [11].

MicroRNA is an approximately 22-codon-long ribonucleic acid molecule created from larger hairpin precursors [70]. It is estimated that around 60% of all protein-coding genes are regulated by miRNAs, whereas one miRNA targets hundreds of genes [5].

Filipowicz et al. stated that miRNAs interact with the corresponding mRNA by base pairing, yet it pairs imperfectly. The miRNAs interact with mRNA mostly in the post-translational phases. The possible mechanisms of the miRNA-mediated post-translational impact on mRNA are repressing deacetylation (before decapping and degradation), repressing cap recognition or 60 s joining, and proteolysis of nascent peptide or blocking elongation by slowing it or dropping-off [70]. MicroRNA seems to have different functions in the cancer microenvironment. Some of these regulate the epigenetic processes by direct modulation of the enzymes taking part in methylation-mediated silencing and chromatic modeling or by epigenetically regulating themselves. MicroRNA also regulates RNA via exosomes, microvesicles, and protein complexes in a paracrine manner to influence the tumor microenvironment. MicroRNA also promotes the release of mediators in both pro- and anticarcinogenesis directions [77]. MicroRNA is classically perceived as either an oncogenic or tumor suppressor. Tumor suppressor means targeting oncogenic particles and molecules; therefore, the normal states combined with its function dictate the end phenotype [77].

Long non-coding RNAs (lncRNAs) are non-coding transcripts shorter or equal to over 200 nucleotides. They take a significant part in many molecular mechanisms like transcriptional and post-transcriptional regulation, processing different types of ncRNAs. LncRNA is responsible for the inactivation of X-chromosome and takes part in cell differentiation, immune responses, apoptosis, and T cells activation. In cancer, the role of lncRNA is complex and includes sustaining proliferative signaling, equivocating growth suppressors, enhancing replicative immortality, creating metastases, enhancing angiogenesis, resisting cell death, and reprogramming the energy metabolic paths [11]. LncRNA can be divided into prooncogenic and tumor-suppressing. Some examples of the prooncogenic long noncoding RNA are PCGEM1, HOTAIR, MALAT1, and CDK2B-ASI. Representative examples of the second group are GAS5, TERRA, and CCND1 [11].

In the table below (Table 3), we summarize the miRNA and lncRNA that bind to particular genes and influence their input in the progression and inhibition of pancreatic adenocarcinoma.

Table 3 presents some of the researched miRNA and lncRNA that play a significant role in the PDAC, as well as its up- or down-regulation status and its pro- or anticarcinogenic characteristics.

From the above-stated theses, it is known that both miRNA and lncRNA interfere with mRNA, causing tumor enhancement or attenuation. It seems that more research is needed in the field of miRNA than lncRNA in PDAC and associated cancerous processes. Yet, we do not know all mechanisms behind those changes and ways to use them in order to stop cancer progression.

## 7. Cross-Talk between Chosen ncRNA Types and Microbiome in PDAC 

The human microbiota is both a collection of microbes colonizing different sites of the human body and its ecosystem [153]. The intestinal tract microbiota is a combination of over 10^14^ microorganisms [153], whilst the microbiome is the genetic information encoded by microbiota (bacteria, fungi, viruses, and archaea) and its host [153].

The microbiome of a healthy pancreas differentiates from the microbiome in a cancerous pancreas (later referred to as “cancer microbiome”) with their composition and ratios of different species of bacteria and viruses [9]. Moreover, the “healthy” microbiota consists mostly of bacteria belonging to *Firmicutes*, *Bacteroidetes*, *Actinetobacteria*, and *Proteobacteria* [1], whereas the microbiota of a cancerous pancreas consists mostly of *Proteobacteria*, *Bacteroides*, and *Firmicutes* according to Pushalkar et al. [9]. It is known that mRNA can enter bacteria via endocytosis [154].

Figure 2 presents the schematic composition of pancreatic microbiota in eubiosis and a microbiome in the state of carcinogenesis. The most common species of bacteria from the most to the least numerous are listed. Additionally, it is important to mention that the number of viruses increases in the cancer microbiome. It is based on the works of Daniluk et al. 2021 and Gesualdo et al. 2020 [1,155].

Riquelme et al. stated that the immune microenvironment and the immune responses are created by the crosstalk between a tumor and its microbiome [156]. Moreover, tumor-associated specific microbiota is expected to directly regulate cancer initiation, progression, and responses to chemo- or immune-therapies [157]. Microbiota can enhance tumorigenesis through epigenetic effects, regulation of miRNA expression, inflammation induction, DNA damage, and differential expression of driver genes [158].

The two strongest theories about its mechanism for altering cancer are that gut microbes release metabolites that affect cancer cells [7], and that the microbes suppress the immune system by creating inflammation [1]. Daniluk et al. found that the Gram-negative bacteria component, namely lipopolysaccharide, enhances the oncogenic signaling by promoting pancreatic inflammation [1]. It is possible through activation of Toll-like receptors, and increasing the tumor-promoting myeloid-derived suppressor cells (MDSC) as well as M2 macrophages [9]. Work by Yuan et al. sheds light on the miRNA regulating the microbiome; they discovered that the profusion of *Akkermansia* correlates with miRNA differentially expressed in samples with colorectal cancer and intestinal tissue around cancer [159].

The report of Wang et al. showed that miRNA and particular bacteria in the gut microbiome co-influence each other, which can be the cause of carcinogenesis in the pancreas [7]. Additionally, it suggested the possibility of using of the combined impact of the microbiome and noncoding RNA to suppress the development of PDAC.

Increased amounts of *Herbaspirillum* and *Catenibacterium* are correlated with NRAS and PT53 mutations. Moreover, a decreased amount of *Barnesiella* is associated with mutated RAS genes [160].

*Fusobacterium nucleatum* bacteria in colorectal cancer may be influenced by an elevated glycan production associated with miRNA [159]. This shows a potential mechanism of miRNAs to modify the microbial composition by regulating glucose metabolism [7]. Marin-Muller et al. found that MiR-198 acted as a crucial tumor suppressor that modulates the molecular composition of an essential interactome in pancreatic cancer [161]. Additionally, they explored a reciprocal regulatory loop between mesothelin and miR-198, which partially explains the MSLN (mesothelin)-gene-mediated pathogenesis [161]. Gironella et al. proved that in mouse xenograft models of pancreatic cancer (TP53INP1), a protein involved in the proapoptotic response upon p53 activation is down-regulated by miR-155 [120].

Shirazi et al. stated that H. pylori and P. gingivalis play a significant role in altering the cell cycle, damaging DNA, influencing the expression of miRNA, and affecting epigenetics in pancreatic cancer [162].

On the other hand, Guo et al. stated that host genetics generally influence microbiota in pancreatic cancer.

Pushalkar et al. shed new light on the influence of non-coding RNA on the microbiota by stating that the mutated KRAS gene may influence the diversity and composition of pancreatic and gut microbiota [9].

So far, the most researched area of the relationship between microbiota and miRNA in neoplasm is colorectal cancer [163]. Malmuthuge et al. suggested that microbiota-associated oncogenic miRNAs could be used as a way to treat cancers both locally and systemically by targeting microbial interventions [164]. Numerous researchers have informed about an urgent need for further research into the relationship between microbiota and non-coding RNA in pancreatic cancer as it may revolutionize the way we perceive oncogene regulation and creation of metastases [165,166].

Moreover, Sammallahti et at. stated that bacterial metabolites are expected to influence both the tumorigenic processes and cancer mutations themselves [158].

Additionally, the possible genetic toxicity and further interference with DNA are another possible mechanism that is used by microbiota to moderate DNA and epigenetic factors accelerating and influencing carcinogenesis. For example, reactive nitrogen species and other toxins are mediated by microbiota [167,168].

The above-mentioned studies suggest that the influence of microbiota on cancer mutations as well as epigenetics is more complex than we have thought.

Many studies have been conducted on the relationship between the microbiota and various cancers, but data on the effect of the microbiome on miRNAs in pancreatic cancer are still lacking. Sammallahti et al. suggested that stool represents a perfect material for researching microbiota and oncogenomic factors such as non-coding RNA, especially for seeking novel biomarkers of pancreatic ductal adenocarcinoma [158]. That opens a new field for further research.

## 8. Conclusions and Knowledge Gaps

The interactions between miRNA, lncRNA, and microbiota are a net of complex yet undefeated connections influencing carcinogenesis. Elinav et al. perceived a desperate need for studies of the microbiome both across the cancer continuum and across cancer types on a greater scale [8]. Therefore, further research in this field can provide us with early-stage diagnostics for pancreatic cancer, alternative targeted therapy options, and possibly a path to pancreatic cancer prevention. As the non-coding RNA field is being widely explored, the aspect of the cross-talk between miRNA and lncRNA and the pancreatic microbiome is still an open field for much-needed research.

Listed below are some of the key aspects of this area that require urgent research. Currently, we do not understand the mechanistic details of the function of miRNAs in repressing protein synthesis. The role of the other components from the microbiota (such as viruses, protozoa, and fungi) in the progression of PDAC and carcinogenesis remains unexplored [8].

Many studies were conducted in order to understand the influences of miRNA on the microbiome and vice versa on the colorectal cancer models, but not yet on pancreatic cancer.

Facing the complexity of the subject, many researchers may be discouraged to conduct experiments in this bidirectional approach, yet many studies suggest that this can lead to disenchanting the severity of PDAC. The co-influence of the microbiome and non-coding RNA creates opportunities to find new points for low-invasive targeted therapy, a gene-blocking approach to treating PDAC, and new biomarkers for diagnostic and treatment evaluation.

A promising application of the interaction of miRNAs and the microbiome would be to administer them orally, paving the way for future miRNA treatment in inflammation or cancer. As novel studies suggest, the implementation of the usage of the co-influence of ncRNA and the microbiome is not only possible but also easy to distribute. Therefore, it is tempting to say that the future will bring a safe and definite cure to one of the deadliest cancers—pancreatic cancer.

## Figures and Tables

**Figure 1 jcm-12-05643-f001:**
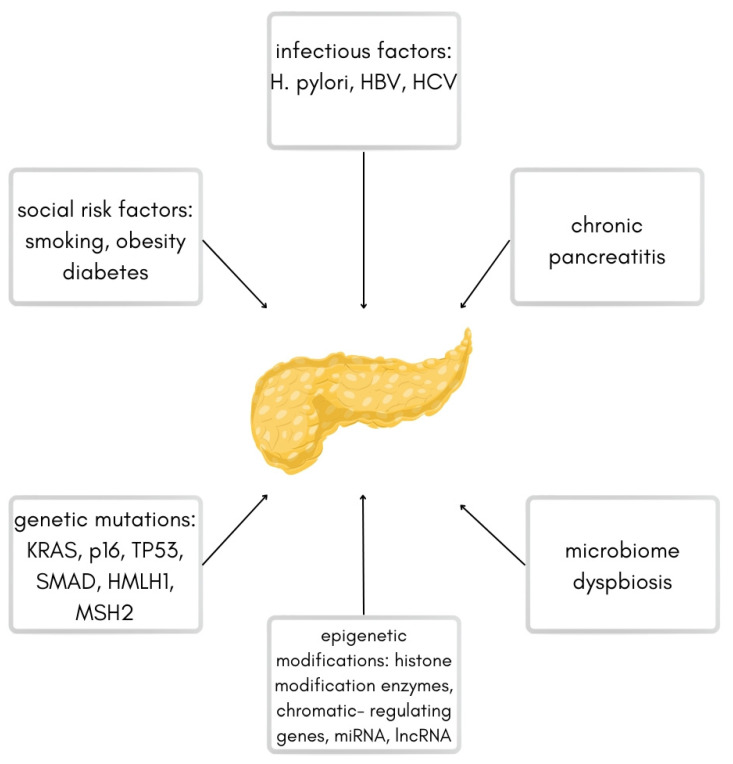
Scheme of different risk factors for pancreatic ductal adenocarcinoma.

**Figure 2 jcm-12-05643-f002:**
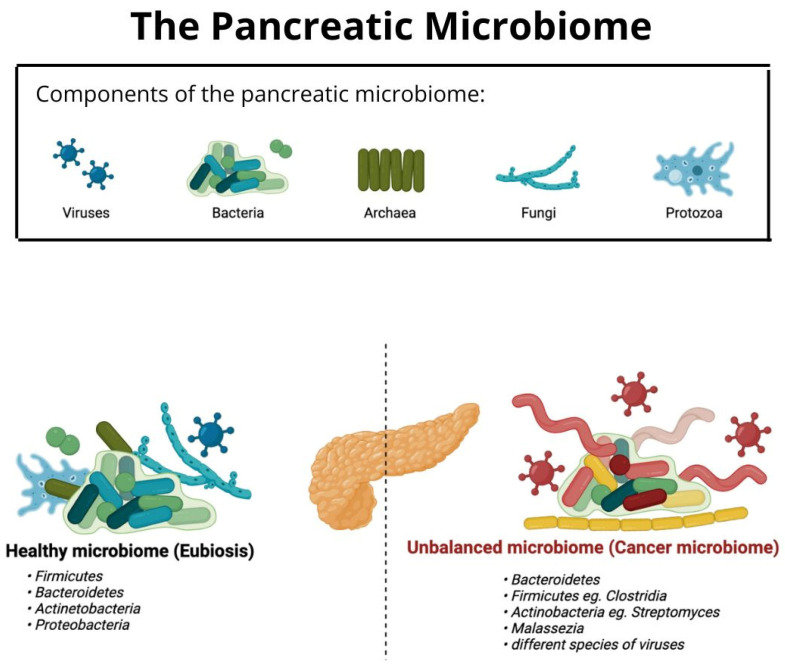
Comparison of the composition of bacterial microbiota of a healthy and cancerous pancreas.

**Table 1 jcm-12-05643-t001:** Genetic mutation in PanIN (pancreatic intraepithelial neoplasia), MCN (mucinous cystic neoplasm), and IPMN (intraductal papillary mucinous neoplasm).

Mutation	PanIN	IPMN	MCN
KRAS codon 12	(+) in 90% of PanINs	(+) KRAS2 most often	(+) in chr.12p from 20% of the least malignant to 89% in most advanced stages
loss of p16 (CDK22A/INK4A) gene on chr. 9p21	(+) in 30% in I, 55% in II, 71% in III	(−)	(−)
tp53 in chr. 17 (inactivation of MADH4/SMAD/DPC4)	(+) in PanIN II and III	(+) only in invasive component	(+) only in invasive component
BRCA2 (ch.19q)	(+) in PanIN II and III	(−)	(−)
mucins (MUC-1, MUC-2,MUC3)	MUC-1	MUC-1, MUC-2, MUC-3	(+) in different glycoforms
STK11/LKB1	(−)	(+) in 30% of lesions	(−)
RNF43	(−)	(+) in 15% of lesions	(+)
PI3KCA	(−)	(+)	(−)
GNAS	(−)	(+) in 4% of lesions	(−)

**Table 2 jcm-12-05643-t002:** Map of genetic alterations associated with various premalignant lesions and PDAC.

Mutation	IPMN	PanIn I	PanIn II	PanIn III	MCN	PDAC	Metastasis
KRAS	+	+	+	+	+	+	+
GNAS	+	−	−	−	−	+	−
Telomere disintegrity	−	+	+	+	−	−	−
MUC1, MUC2, MUC3	−	+	+	−	−	−	−
CDK2	−	+	+	+	+	+	−
BRCA2	−	+	+	−	−	−	−
RFN43	−	−	−	−	+	+	−
PI3KCA	−	−	+	−	−	−	−
STK11/LKB1SSTK11/LB1	−	−	+	−	−	−	−
SMAD4	+	−	−	+	+	+	+
CDKN2A	+	−	−	−	−	+	−
tp53	−	−	−	−	+	+	+
FOXA1	−	−	−	−	+	−	−

**Table 3 jcm-12-05643-t003:** microRNA and long non-coding RNA and their genetic targets in PDAC.

miRNA	Up/Down-Regulated in PDAC	Targeted Genes	Pro/Anticancer Outcome	Description
miR-107	up	CDK6 [78]	pro	stops growth
miR-127	down	BAG5 [79]	anti	inhibits cancer development
miR-21	up	Bcl-2, FasL [80]	pro	decreases apoptosis, increases gemcitabine resistance
		FoxO1 [81]	pro	increases tumor growth
		PDCD4 [82]	pro	increases proliferation and reduces cell death
		PTEN, RECK [83]		intensifies the progression of cell cycle, increases proliferation
miR-221	up	p27kip1 [83]	pro	enhances the progression of cell cycle, promotes proliferation
		PTEN, P27KIP1, P57KIP2, PUMA [84]	pro	increases proliferation
		TIMP2 [85]	pro	increases proliferation and invasion, stops apoptosis
		TRPS1 [86]	pro	Mediates EMT phenotype, migration, and growth
miR-222	up	p57 [87]	pro	enhances proliferation
		MMP2, MMP9 [85]	pro	enhances proliferation, invasion, stops apoptosis
miR-10 a	up	HOXA1 [88]	pro	enhances invasion
miR-125 b	up	TXNIP [89]	pro	enhances tumorigenesis and progression
miR-34 a	down	CD133, Notch1, Notch2, Notch4 receptors [90]	anti	inhibits cell survival, invasion, migration, increases cell apoptosis
miR-143	down	KRAS [91]	anti	inhibits cell proliferation, migration, and invasion
miR-145	up	KRAS, RREB1 [92]	anti	stops tumor growth
		ROR [93]	anti	stops proliferation, invasion, and cell cycle
		MUC13 [94]	anti	stops tumor growth and invasion
miR-217	down	SIRT1 [95]		regulates (epithelial–mesenchymal transition) EMT process
		KRAS [96]	anti	stops cell growth and colony forming
miR-141	down	MAP4K4 [97]	anti	stops proliferation, colony formation, invasion by inhibiting G1-phase, and apoptosis
		YAP [98]	anti	stops proliferation, forming colonies, and apoptosis
		TM4SF1 [99]	anti	stops invasion and migration
miR-148 a	down	DNMT1 [100]	anti	stops proliferation and metastasis
		CCKBR, Bcl-2 [101]	anti	stops tumor growth and apoptosis
		CDC25B [102]	anti	stops cell survival
miR-375	down	PDK1 [103]	anti	stop cell growth and enhances apoptosis
miR-29 c	down	ITGB1 [104]	anti	stops cell growth, invasion, and migration
		MMP2 [105]	anti	stops migration, invasion, metastasis (in mice model)
		FRAT2, LRP6, FZD4, FZD5 [106]	anti	stops migration and stem-cell-like phenotype
miR-130 b	down	STAT3 [106]	anti/pro	stops invasion and encourages proliferation
miR-200 c	down	MUC4, MUC6 [107]	anti	targets directly MUC4 and 6
		E-cadherin [108]	anti/pro	stops invasion and enhances proliferation
miR-216 a	down	JAK2 [109]	anti	stop proliferation and enhances apoptosis
		Beclin-1 [110]	anti	enhances radiosensitivity
miR-26 a	down	p53 [111]	anti	stops proliferation by phosphorylation of p53
		Cyclin E2 [112]	anti	stops proliferation
miR-148 b	down	AMPKalfa1 [113]	anti	stops cell cycle and cell growth
		DNMT1 [114]	anti	influences methylation of tumor suppressor genes
miR-335	down	OCT4 [114]	anti	stops progression and influences stem cell properties
miR-365	down	SHC1-BAX [115]	pro	elicits gemcitabine resistance
miR-155	up	Foxo3a, KRAS, ROS [116]	anti	decreases proliferation induced by ROS generation
		SEL1L [117]		downregulates SEL1L
		MLH1 [118]		downregulates MLH1
		SOCS1 [119]		enhances invasion and migration
	up	TP53INP1 [120]		enhances tumor growth
miR-23 a	up	APAF1 [121]	pro	increases proliferation and decreases apoptosis
		FZD5, HNF1B, TMEM92 [122]	pro	increases EMT-like cell transformation
miR-143	up	ARHGEF1, ARHGEF2, KRAS [123]	anti	decreases migration, invasion, and metastasis to liver
miR-146 a	up	EGFR, IRAK1, MTA-2 [124]	anti	stops invasion
miR-150	up	MUC4 [125]	anti	stops growth, clonogenicity, migration, invasion, enhances intercellular adhesion
miR-181 a	up	PTEN, MAP2K4 [126]	pro	enhances migration
		TNFAIP1 [127]	pro	enhances proliferation and migration
miR-214	up	ING4 [128]	pro	decreases sensitivity to gemcitabine
miR-15 b	up	SMURF2 [129]	pro	enhances EMT
miR-23 b	up	ATG12 [130]	pro/anti	regulates autophagy associated with radioresistance
miR-24	up	Bim [131]	pro	increases cell growth
		FZD5, HNF1B, TMEM92 [122]	pro	increases EMT-like cell share transformation
miR-92 a	up	DUSP10 [132]	pro	enhances proliferation
miR-181 b	up	BCL-2 [133]	anti	sensitizes particular cells to gemcitabine
		CYLD [134]	anti	increases gemcitabine resistance of some cells
miR-196 a	up	NFKBIA [135]	pro	enhances proliferation and migration
		ING5 [136]	pro	enhances proliferation, migration, and decreases apoptosis
miR-27 a	up	Sprouty2 [137]	pro	enhances growth, colony formation, and migration
miR-223	up	FBw7 [138]	pro	secures EMT phenotype
miR-320 a	down	DGCR5 (lncRNA) [139]	anti	regulates proliferation, migration, and 5-FU resistance
miR-31	up	APBB2 [140]	anti	reduces migration of cancer cells
miR-451	up	CAB39 [141]	pro	enhances cell proliferation and lymphatic metastasis
miR-let7	down	numerous genes in the insulin signaling pathway [142]	anti	inhibits tumor progression and increases therapy sensitivity
miR-100		FGFR3 [143]	anti	inhibits proliferation and enhances sensitivity to cisplatin
**lncRNA**	**Up/Down-Regulated in PDAC**	**Targeted Genes**	**Pro/Anticancer Outcome**	**Description**
AF339813	up	NUF2, CDK1, CDK4/CDK6 [14]	pro	apoptosis, controls cell cycle
AFAPI1-AS1	up	E-cadherin, N-adherin, Snail [14,144]	pro	regulates cell proliferation, migration, invasion
BC008363	down	many protein-coding genes involved in tumor growth and drug resistance [14,145]	anti	diminishes tumor growth and rug resistance
CDKN2B-ASI	up	miRN-411-3p [146]	pro	regulates miRN-411-3p and HIF-1alfa (hypoxia-inducible factor)
ENST00000480739	down	OS-9, HIF-1 [14]	pro	controls invasion and migration
GAS5	up	miR-32-5p [147]	anti	regulates the cell cycle, involved in PI3K/Akt signaling pathway
GAS5	down	CDK6 [14]	ani	stops cell proliferation
H19	up	Let-7, HMGA2, miR-194, miR-138, miR-200 [14,148]	pro	mesenchymal–epithelial transition (MET), involved in development and progression
HOTAIR	up	PRC2, GDF15 [14]	pro	related to invasion, proliferation, progression of PC
HOTTIP	up	AURKA, WDR5, HOXA10, HOXB2, HOXA11, HOXA9, HOXA1, HOXA13 [14]	pro	cell cancer proliferation, stops cell apoptosis, enhances migration of cancer cells
MALAT-1	up	Sox2, E-cadherin, N-adherin, vimentin, VEGF [14]	pro	regulates cell cycle, growth, migration, and invasion
PCGEM1	up in most cancers	numerous miRNA [149,150]	pro	involved in cell invasion and migration, associated with cancer progression
PLACT1	up	hnRNPA1 [151]	pro	regulates tumorigenesis through NF-kappa beta signaling pathway, involved in lung metastasis progression
TERRA	up	TRF2 [152]	anti	induces apoptosis, inhibits cell proliferation, invasion, metastasis, regulates cell cycle

## Data Availability

No new data were created or analyzed in this study. Data sharing is not applicable to this article.

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
