# Peer review of "Microbiome and MicroRNA or Long Non-Coding RNA—Two Modern Approaches to Understanding Pancreatic Ductal Adenocarcinoma"

_jcm, 2023, doi:10.3390/jcm12175643_

Round 1
Reviewer 1 Report
This review paper provides a comprehensive overview of recent advancements in understanding the molecular mechanisms underlying pancreatic carcinogenesis. At first, this report explains the complexity of pancreatic cancer and its origins, emphasizing the significance of conducting omics research on the non-resective cases of pancreatic adenocarcinomas, such as analyzing biopsy material. Including detailed figures and tables enhances the reader's understanding, showcasing common genetic mutations at different stages of pancreatic ductal adenocarcinoma (PDAC) and subsequent metastases. The review also highlights characteristic mutations present in pre-malignant lesions observed before the development of PDAC. In conclusion, the authors emphasize the need for further exploration into the crosstalk between microRNAs (miRNA) and long non-coding RNAs (lncRNA) and the interactions within the pancreatic microbiome. These insights hold promise for future applications, particularly the potential of miRNA-based treatments for inflammation or cancer.
Overall, the review is well-constructed and effectively conveys complex information. However, it is essential to note that there are a few inaccuracies in the references. For example, on page 1, line 29, the correct citation is Wang et al., not Feng et al. Similarly, on page 1, line 42, the reference to Guruswamy et al.'s work is mentioned but not listed correctly in the reference section.
Author Response
Wiktoria Izdebska
Department of Gastroenterology and Internal Medicine,
Medical University of Bialystok,
Bialystok 15-276
Poland
18.08.2023
Dear Editors and Reviewers,
We thank you for taking into consideration our manuscript entitled “Microbiome and miRNA or lncRNA- two modern approaches to understanding pancreatic ductal adenocarcinoma” to be published in the Journal of Clinical Medicine. We have reviewed the comments and considered them very carefully. The corrections in the manuscript are in blue color as well as they are marked by a comment in a Microsoft Word file. The point-by-point responses to reviewers’ comments are presented below. We thank the reviewers for their insightful comments during the submission process.
Sincerely,
Wiktoria Izdebska
REVIEWER 1:
Reviewer’s comment #1: there are a few inaccuracies in the references, such as on page 1, line 29, the correct citation is Wang et al., not Feng et al., on page 1, line 42, the reference to Guruswamy et al.'s work is mentioned but not listed correctly in the reference section.
Answer: All above stated inaccuracies have been corrected. Additionally, we have revised the references in order to correct possible mistakes.
REVIEWER 2:
Reviewer’s comment to major point #1: It is recommended to reduce the amount of content dedicated to genetic alterations. A brief overview of key genetic factors related to PDAC will suffice, emphasizing their relevance to the interactions between the microbiome and miRNA or lncRNA in the disease context.
Answer:
We thank you for drawing our attention to this aspect, as it is true that a significant but not necessary amount of information is about mutations causing pancreatic ductal adenocarcinoma. We have shortened the third and fourth chapters dedicated to genetic alterations.
Reviewer’s comment to major point #2: One aspect that requires further elaboration is the specific interactions between the microbiome and miRNA or lncRNA in the context of PDAC. The paper briefly mentions this relationship, but it would greatly benefit from a more in-depth exploration of the mechanisms through which the microbiome influences the expression and regulation of miRNA or lncRNA in PDAC, and vice versa.
Answer:
We thank you for drawing our attention on this aspect of our work. We supplemented our manuscript with further essential information on the co-influence of microRNA and pancreatic microbiota. All changes are equipped with comments in order to easily find them in the chapter no. 7.
Reviewer’s comment to minor point #1, 2, 3:
#1 The phrase that “ The innovative 10 report of Wang et al. showed that miRNA not only affects particular bacteria in the gut microbiome which can promote carcinogenesis in the pancreas but also the microbiome has a visible impact on the miRNA” are repeatedly used in the same manuscript.
#2 Page 3, Lines 92-94: There are some mistakes for citation. For example, “… physiological pocesses22)14,23” seems strange in numerous places in the text.
#3 Page 7: In Figure 2, there is a possibility of misleading the readers. I believe that PanIN, IPMN, and MCN are each precancerous lesions of PDAC. However, I do not think that they progress in the sequence IPMN → PanIN → MCN. Nevertheless, the Figure may give the impression that these three lesions form a single continuous progression.
Answer: All above stated inaccuracies have been corrected and repeated sentences are rephrased. Additionally, we have revised English in the entire text in order to improve readability of the manuscript and its core matter. The figure 2 has been changed to table 2 in order to not mislead the readers.
Reviewer 2 Report
I would like to express my gratitude for the opportunity to review the paper titled "Microbiome and miRNA or lncRNA - two modern approaches to understanding pancreatic ductal adenocarcinoma" by Dr. Wiktoria Maria Izdebska. The paper presents valuable insights into recent findings of microbiome and miRNA or lncRNA in the field of pancreatic ductal adenocarcinoma (PDAC).
Overall, the paper is well-written and insightful; however, I have a couple of concerns that I believe could improve the manuscript.
Major points
# The paper dedicates a significant portion to discussing genetic alterations in PDAC and precancerous lesions. While this information is undoubtedly relevant to the broader understanding of the disease, it appears to be slightly disconnected from the main focus of the study. To enhance the clarity and conciseness of your work, I recommend reducing the amount of content dedicated to genetic alterations. A brief overview of key genetic factors related to PDAC will suffice, emphasizing their relevance to the interactions between the microbiome and miRNA or lncRNA in the disease context.
# One aspect that requires further elaboration is the specific interactions between the microbiome and miRNA or lncRNA in the context of PDAC. The paper briefly mentions this relationship, but it would greatly benefit from a more in-depth exploration of the mechanisms through which the microbiome influences the expression and regulation of miRNA or lncRNA in PDAC, and vice versa. Providing concrete examples and experimental evidence, where available, will strengthen your arguments and help readers grasp the significance of these interactions.
Minor points
Abstract
# Page 1, Lines 10-13: The phrase that “ The innovative 10 report of Wang et al. showed that miRNA not only affects particular bacteria in the gut microbiome which can promote carcinogenesis in the pancreas but also the microbiome has a visible impact on the miRNA” are repeatedly used in the same manuscript, which seems repetitive and tedious for the reader. Please paraphrase this as much as possible.
# Page 3, Lines 92-94: There are some mistakes for citation. For example, “… physiological pocesses22)14,23” seems strange to me. I saw several parts like this. Please correct them.
# Page 7: In Figure 2, there is a possibility of misleading the readers. I believe that PanIN, IPMN, and MCN are each precancerous lesions of PDAC. However, I do not think that they progress in the sequence IPMN → PanIN → MCN. Nevertheless, the Figure may give the impression that these three lesions form a single continuous progression.
English quality is good and only a few corrections are needed.
Author Response

(The authors gave the same response as above.)
